# Morton’s Neuroma or Intermetatarsal Bursitis—A Prospective Diagnostic Study of Intermetatarsal Pain

**DOI:** 10.3390/diagnostics15111339

**Published:** 2025-05-26

**Authors:** Sif Binder Larsen, Cecilie Mørck Offersen, Eva Dyrberg, Jens Kurt Johansen, Naja Bjørslev Lange, Birthe Højlund Bech, Michael Bachmann Nielsen, Søren Tobias Torp-Pedersen

**Affiliations:** 1Department of Diagnostic Radiology, Rigshospitalet, 2100 Copenhagen, Denmark; 2Department of Clinical Medicine, University of Copenhagen, 2200 Copenhagen, Denmark; 3Department of Orthopedic Surgery, Hvidovre Hospital, 2650 Hvidovre, Denmark; 4Department of Orthopedic Surgery, Bispebjerg and Frederiksberg Hospital, 2400 Copenhagen, Denmark

**Keywords:** Morton’s neuroma, intermetatarsal bursitis/bursa, web space, ultrasound, magnetic resonance imaging, diagnostic criteria, prospective study

## Abstract

**Background:** Intermetatarsal bursitis (IMB) is emerging as a diagnostic consideration for patients with forefoot pain. However, few investigations have been conducted into the incidence of IMB among patients with forefoot pain. The symptoms of IMB are described as mimicking those of Morton’s neuroma (MN). Currently, the best method to differentiate between MN and IMB is radiological evaluation. Based on this, the aim of this study was to investigate the incidence of IMB and MN in a prospective cohort of patients with intermetatarsal pain diagnosed with radiological evaluation and compared to a control group. **Methods:** This study included 26 patients and 13 controls. All participants underwent magnetic resonance imaging (MRI) and ultrasound (US) of one forefoot. **Results:** Among the 26 patients, 5 (19.2%) had MN and 14 (53.8%) had IMB on MRI compared to US, with which 25 (96.2%) cases of IMB and 0 with MN were identified. In the control group, both modalities found asymptomatic web space pathology in four cases (30.8%), and US identified normal intermetatarsal bursas in five cases. Additionally, our results indicate that MN patients have more severe pain and a longer history of pain compared to IMB patients. **Conclusions:** Based on our MRI results, we conclude that IMB is frequent in patients with intermetatarsal pain. Differentiation between MN and IMB with US is complex and should be performed with caution and an understanding of both conditions. Normal intermetatarsal bursas are also visible on US as hypoechoic but non-expansive masses.

## 1. Introduction

Pain in a forefoot web space is a common cause of metatarsalgia and can limit the patients’ ability to participate in their everyday activities [1]. Pathologies in this area include Morton’s neuroma (MN), a focal benign thickening of the plantar digital nerve near its separation into branches for the toes, and intermetatarsal bursitis (IMB), which is pain caused by inflammation in an intermetatarsal bursa located between the metatarsal heads above the deep transverse metatarsal ligament (DTML); in web spaces two and three, it extends distally past the metatarsophalangeal joints [2,3]. The two structures lie in close proximity, and the shared anatomic location complicates differentiation during clinical examination [4]. The symptoms and clinical findings of MN include burning or shooting pain, altered sensation, and web space pain upon palpation [5,6]. The clinical features of IMB have yet to be systematically investigated, but IMB is described as a cause of metatarsalgia, and improvement in symptoms after treatment with insoles has been demonstrated [7,8].

Although IMB is often mentioned as a differential diagnosis to MN [9,10], its incidence and clinical significance remain poorly understood due to limited studies on the condition among patients with forefoot pain [11,12]. If IMB is a differential diagnosis to consider for patients with web-space-related forefoot pain, it should be expected to appear in imaging assessments of these patients. Despite the expected appearance, there is a notable gap in the reporting of IMB cases, especially in the literature about MN.

To investigate the clinical significance of IMB, the first requirement is to be able to diagnose IMB and differentiate it from MN. Currently, the best described method to differentiate between the two conditions is diagnostic imaging, which is frequently used to support the evaluation of forefoot pain. When assessing the web space, ultrasound (US) and magnetic resonance imaging (MRI) are the recommended modalities [13,14,15]. Patients with MN frequently experience delayed diagnosis and variable treatment outcomes [16,17,18]. A deeper understanding of the imaging characteristics, the underlying causes, and the relationship between MN and IMB could enhance diagnostic accuracy and improve patient management, ultimately reducing pain duration and optimizing treatment strategies.

Based on this, we hypothesize that some patients currently diagnosed with MN based on clinical examination have IMB. Our study aimed to investigate the incidence of IMB and MN based on radiological evaluation of the forefoot using US and MRI in a group of patients with pain in an intermetatarsal space compared with a control group of subjects without forefoot pain. The primary endpoint was the incidence of IMB and MN in both subject groups. The secondary endpoint comprised differences in symptoms and clinical parameters between patients with MN and IMB. In this paper, we present the results of our study, the observed benefits and limitations of diagnostic imaging in this setting, and the implications of our findings.

## 2. Materials and Methods

In this prospective diagnostic study, we used MRI and US to evaluate intermetatarsal spaces in a patient group with web space pain and a control group without forefoot pain. Patients were referred to our study from orthopedic departments within the Capital Region in Denmark. Scans were obtained at Rigshospitalet.

The study was approved by the Ethics Committee of the Capital Region of Denmark (approval number: H-20067346) and the Center for Data Registration (approval number: P-2020-835). Informed and written consent for participation was obtained from all participants involved in the study. The study was registered at www.clinicaltrials.gov (accessed on 12 June 2022), registration number: NCT05685160.

**Participants:** Study participants were consecutively recruited from July 2021 to September 2023. Patients were referred to our study from orthopedic departments in the Capital Region of Denmark. To be assessed for eligibility, all patients had to be examined by an orthopedic surgeon who found pain in at least one intermetatarsal space, and MN or IMB was the primary diagnostic consideration. Controls were recruited through the online clinical study recruitment site www.forsøgsperson.dk (accessed on 28 October 2022), which allowed all interested individuals to contact us. We also placed recruitment posters in the waiting areas of our departments, but this did not result in any inclusions.

Participants were included if they were >18 years old and an orthopedic surgeon found web space pain at clinical evaluation (patients only). The exclusion criteria were (1) ongoing infections or wounds in the foot at the time of the scans; (2) contraindications to participating in the MRI scan; (3) previous treatment for MN or IMB, i.e., surgery at any time or corticosteroid injection within the past 6 months; (4) a history of significant trauma in the forefoot, e.g., fracture or previous surgery; (5) impaired kidney function (eGFR < 30 mL/min); and (6) pain in the foot (controls only).

Participants were included if they met the inclusion criteria and none of the exclusion criteria. The following clinical information for the patient group was collected: symptoms, affected foot, subjective pain localization, change in sensation of the adjacent toes, pain duration, VAS score, triggering trauma, and the Self-Reported Foot and Ankle Score (SEFAS) questionnaire [19]. The pain localization assessed by an orthopedic surgeon was also collected. If both feet were affected, the foot with the most severe symptoms (according to the patient) was included in the study.

**Ultrasound evaluation:** A senior radiologist with 30+ years of experience in musculoskeletal ultrasound performed the US evaluation. The US assessment was performed using a LOGIQ E10 ultrasound scanner with a 6–15 MHz linear and 6–24 MHz hockey stick transducer (GE HealthCare, Chicago, IL, USA). The scan was obtained from a dorsal approach with the participant’s foot placed flat on the examination bed. The forefoot was assessed systematically from the medial to the lateral side, evaluating all metatarsophalangeal joints and intermetatarsal spaces. The participants were instructed not to engage in conversation with the radiologist until after the scan and only to agree on the foot of interest. This approach was applied to minimize the risk of revealing to which group (patient/control) they belonged. The diagnostic criteria for MN on US were a hypoechoic mass between the metatarsal heads closest to the plantar surface [4,20]. The interdigital nerve may be seen entering the mass [21]. For IMB, the criteria were a rounded hypoechoic mass between metatarsal heads closest to the dorsal surface with possible activity on color Doppler [22]. If the mass appeared non-expansive with concave areas, it was considered a normal intermetatarsal bursa.

**MRI evaluation:** The MRI scans were performed on a 3T SIGNA Premier (GE HealthCare, Chicago, IL, USA). The MRI scan was assessed by two senior radiologists with 30+ and 10 years of experience. In cases of doubt, consensus was reached by consulting a third reviewer, who was a senior musculoskeletal MRI radiologist. The participants were scanned in a prone position as recommended by Zanetti and Weishaupt 2005 [10]. The imaging protocol included coronal STIR, axial T1 FSE, coronal T1 FSE, axial T2 FSE, and coronal and axial T1 FSE Flex (Dixon method) with contrast. Gadovist 0.1 mL/kg was used as the contrast agent (Bayer AG, Berlin, Germany). We chose to include contrast-enhanced sequences because of our focus on distinguishing IMB from MN [10,23]. The imaging criteria for MN were a demarcated process between the metatarsal heads with low to intermediate signal on T1 and T2, isointense to muscle, and showing contrast enhancement in post-contrast sequences [24]. IMB was diagnosed when a process between the metatarsal heads showed low signal on T1, high signal on T2, and peripheral enhancement was observed in post-contrast sequences [23]. The method of differentiation with MRI is visually summarized in Figure 1.

For each modality, the results were grouped into four categories: (1) IMB, if IMB was reported; (2) MN, if any MN was reported; (3) Normal scan, for scans without reported pathologies; and (4) Other, for scans with other pathologies than IMB or MN. A subcategory of visible but normal intermetatarsal bursas was added after the data collection. The study was conducted with the US and MRI radiologists semi-blinded to participant type, as the first half of the study was not blinded and only patients were assessed, and the latter half was fully blinded. The controls were included consecutively from October 2022.

**Statistics and data:** The statistical analysis was performed using R (version 2024.04.2). The main outcomes are reported in % (*n*/total of patients or controls). Continuous variables are reported as mean ± standard deviation, unless otherwise specified. The Welch Two-Sample *t*-test was used to assess continuous data (VAS, duration of pain). Fisher’s exact test was used for binary data (symptoms). We considered *p* < 0.05 significant. Pairwise deletion was used in cases of missing data in the subgroup analysis. Study data were collected and managed using REDCap (Research Electronic Data Capture, version 14.5.17) electronic data capture tools hosted by the Capital Region of Denmark (Hillerød, Denmark) [25,26].

## 3. Results

Brief summary of results:Among patients with intermetatarsal pain, 53.8% had IMB and 19.2% had MN on MRI;US identified intermetatarsal pathology in all but one patient, but the differentiation is challenging;A normal intermetatarsal bursa is visible on US;A hypoechoic process between the metatarsal heads on US is not always pathological and should always be interpreted in correlation with the symptoms;Patients with MN have more severe pain compared to patients with IMB;There are no differences in the symptoms of MN and IMB.

### 3.1. Participant Characteristics

Thirty-five patients were referred from orthopedic departments, and eighteen controls inquired regarding participation. Seven patients and five controls were not enrolled as they changed their minds about participation; the reasons are specified in Figure 2. In total, 28 patients and 13 controls were included during the study period. Two patient cases were excluded from the analysis as one did not have both scans performed, as the findings on MRI needed immediate treatment, and in the case of the other, the scans were obtained using the wrong protocol.

Twenty-six patients and thirteen controls were included in the analysis. The mean age was 49.6 (range 27–76) for the patients and 49.45 (range 23–70) for the controls (*p* = 0.9613). The female/male (% female) distribution was 21/5 (80.8%) in the patient group and 9/4 (69.2%) in the control group (*p* = 0.4645). The clinical characteristics of the patients included in the study are presented in Table 1.

### 3.2. Imaging Results

Intermetatarsal pathology in the form of IMB or MN was found in the majority of our patients, with IMB being the most frequent cause of intermetatarsal pain, but there was a notable disagreement regarding the diagnosis between the two modalities. The results are presented in Table 2. Two patients had other pathologies on their MRI scan, which were degenerative changes (*n* = 1) and tenosynovitis (*n* = 1). MRI and US agreed in the case of the normal scan, but with MRI, there were an additional four scans that showed no explanation for the patients’ pain. In the differentiation between MN and IMB, we paid close attention to the anatomic location of the process, as IMB had a more dorsal extent compared to MN, and the two conditions are separated by the DTML at the level of the metatarsal heads. Both conditions had a distal extent past the metatarsophalangeal joints into the space between the proximal phalanx, as illustrated in Figure 3. Imaging results of both modalities from a patient with MN are shown in Figure 4, and from a patient with IMB in Figure 5. US identified pathologic intermetatarsal spaces in all but one patient, but we were unable to differentiate MN from IMB, as illustrated in Appendix A.

In the control group, both modalities found web space pathology without clinical symptoms in four cases; see Table 2 for an overview of the results. There was agreement between the modalities for only two of the cases with asymptomatic web space pathology. In one case, the pathology was diagnosed as MN with MRI and as IMB with US. In the other case, both modalities reported IMB. In the remaining two cases where US reported IMB, MRI reported no web space pathology. For the two cases where MRI found IMB, the US findings were either normal or the bursa was visible but not pathologic. Examples of the web space pathology reported in the control group are presented in Figure 6. The case of *other findings* for MRI in Table 2 was bone marrow edema without any web space pathology. With US, we identified a visible bursa in web space two and/or three in 9 of the 13 controls (69%). Of these, four (31%) were diagnosed as IMB, but the remaining five were assessed as nonpathological but visible, due to their collapsed/non-expansive appearance; see examples in Figure 7.

The time between MRI and US scans varied from the same day to 211 days. The mean time between the scans for the patient group was 12.1 (±19.4) days, and for the control group it was 69.1 (±60.4) days. There was no significant difference in the time between the scans for the patients with IMB compared to those with MN. Graphs presenting the detailed data are available in the Appendix A.

### 3.3. Differences Between MN and IMB

Based on the MRI diagnosis, we investigated whether MN and IMB showed differences in the symptoms and the clinical parameters: duration of pain and VAS score. The dominant symptoms of the included patients were a sharp or shooting pain that was aggravated when walking and alleviated by rest. There was no statistical difference between the symptoms for patients with IMB compared to patients with MN (see analysis in Appendix A). The frequency of all reported symptoms is presented in Figure 8. The mean duration of pain in months was 45 (±36.2) for patients with MN and 23.2 (±28.6) for patients with IMB (*p* = 0.2706). The mean VAS score for patients with MN (85.2 (±15.5)) was significantly higher compared to IMB patients (61.8 (±23.9)) (*p* = 0.0318). The results are shown in Figure 9.

## 4. Discussion

In this prospective study of patients with intermetatarsal pain, compared with a control group, radiological evaluation identified IMB or MN in most patients. MRI identified IMB in 54% of patients, making it the most frequent finding. MRI failed to identify a radiological explanation for the reported pain more often, but US was unable to differentiate MN from IMB. There were no differences in the symptoms between IMB and MN; however, patients with MN had a longer pain history on average and significantly higher VAS scores than those with IMB. In the control group, asymptomatic web space pathology was common on imaging and present in nearly one-third of the individuals. Additionally, we found that the normal intermetatarsal bursa is visible on US.

IMB was the most frequent cause of intermetatarsal pain among our patients. Three previous studies have reported an incidence of IMB and MN among patients with forefoot pain. The incidence of IMB varied from 21 to 81% and that of MN from 11 to 69% [8,12,27]. Our results are within these ranges, and our combined results strengthen the argument for the presence and clinical significance of IMB in patients with metatarsalgia, especially with intermetatarsal pain. The wide range in incidence rates may reflect variations in inclusion criteria (metatarsalgia vs. neuralgia) and differences in diagnostic criteria, which have been sparsely reported in previous studies. Paying attention to diagnostic details, with the aim of making the methodological assessment of these patients more homogenous, could help narrow these intervals to obtain a true scope of the occurrence of IMB.

The deviation between our MRI and US results is primarily caused by the inability to differentiate MN from IMB on US, the main reason for which is their similar sonographic appearance. The challenge of differentiating the US findings of the web space were addressed in previous studies by Cohen et al. 2016, who found the “hypoechoic heterogeneous mass that is referred to as a Morton neuroma sonographically is really a “neuroma–bursal complex”” (p. 2191) and Read et al. 1999 who “could not clearly separate Morton’s neuroma from associated mass-like mucoid degeneration in the adjacent loose connective tissues” (p. 153) [21,28]. Considering this and our focus on IMB, it might have led to the classification of some neuromas as bursitis on US. Additionally, we were not able to identify the plantar digital nerve in relation to the hypoechoic mass, which could have enhanced the diagnostic confidence of MN [29]. The predefined criteria used in our study to differentiate between MN and IMB with US demonstrate their limitations when applied to the study population. Our US results emphasize the complexity of differentiating between MN and IMB, and therefore, differentiation between MN and IMB must be performed with caution and in consideration of both entities. We suggest using the US diagnosis “neuroma–bursal complex” or similar if the origin of the encountered web space pathology is not obvious.

In the control group, we found that the normal intermetatarsal bursa can be visualized on US and appears non-expansive with concave sides. This distinguishes it from pathological web space masses that typically appear expansive. This contrasts with earlier studies by Koski, 1998, who reported that intermetatarsal bursas were rarely seen with US in the normal foot, or by Redd et al., 1989, who found “no normal hypoechoic structures greater than 3 mm in the intermetatarsal space” (p. 416) [30,31]. These earlier findings continue to influence current ultrasound criteria [8,32]. Our results challenge this prevailing view by showing that the intermetatarsal bursa may be seen in a normal web space. US identified visual intermetatarsal bursas in 69% of our control (including those that were reported as IMB), which is in line with the results of Zanetti et al. 1997, who found visible intermetatarsal bursas in 67% of healthy subjects on MRI, indicating that both modalities are now equally able to detect the presence of a normal intermetatarsal bursa [33]. Our findings could be attributed to the advancements in US equipment and software, allowing the detection of structures previously unrecognized. However, given the demonstrated complexity of US examination of the intermetatarsal space, the presence of a non-expansive hypoechoic mass should always be evaluated alongside the ability to provoke pain from transducer pressure to eliminate false-negative examinations.

We expected a variance in the anatomic location of each entity, but in light of our MRI results showing the shared anatomic location distal to the metatarsal heads, differentiation based on location with US seems unlikely. This is backed up by anatomic studies, which have demonstrated that the intermetatarsal bursa in web spaces two and three distends distal and past the metatarsal heads into the space between the metatarsophalangeal joints and the proximal phalanges, where it comes in close proximity to the neurovascular bundle [3,34,35]. Additionally, it has been demonstrated that the bifurcation of the plantar digital nerve, and thus the most common site of MN, is not under the DTML but distal to the DTML [36]. It was our experience from this study that this structural assessment was complicated by the size of the area of interest, which made the deception on MRI images sensitive to variations in imaging planes and slice thickness, especially for more intermediate cases. With improvements in MRI technology and imaging resolution, the enhancement of the anatomic information will aid these structural considerations in the future.

When addressing IMB and MN, it must be acknowledged that whether they are two separate diseases or represent stages of the same pathological process remains to be determined, as the pathogenesis is not evident [37,38]. The “intermetatarsal bursitis theory” suggests that IMB may contribute to the development of MN [2,38,39]. If they are two independent diseases, there is a possibility for them to occur at the same time. In our study, we attempted to differentiate between MN and IMB using radiological evaluation, under the assumption that they are distinct entities. We did not find any cases where both diseases were present, but there was visible fluid in some intermetatarsal spaces in the MN cases. Our results do not exclude a developmental relationship. Our analysis of pain duration and intensity according to MRI diagnosis showed that patients with IMB on MRI have lower pain levels and a shorter history of pain compared to patients with MN. This could be interpreted as a possible progression of IMB to MN within the affected web space, indicating a potential development over time, as previously proposed by Bossley and Cairney, 1980 [3]. This would support efforts to facilitate early intervention to prevent aggravation. A longitudinal design with repeated imaging over time could provide further insight into this potential progression.

The definitive argument for differentiating between MN and IMB is differences in treatment effects and the targeting of surgical treatment strategies, which have yet to be investigated. Our study showed that with radiological evaluation, MRI was superior in the distinction, due to the distinction in the T2 and contrast-enhanced sequences, but also aided by the anatomic overview that this modality provides, which makes it possible to follow the interdigital masses proximally to locate their anatomic origin above or below the DTML. Based on this, we recommend MRI for cases in which differentiation is pivotal to the treatment strategy.

This study has some strengths and limitations that need to be addressed. Its main strengths are its prospective design and representative patient population, recruited during the diagnostic process, which enhances the clinical relevance of our results. However, several limitations must be acknowledged. First, the full blinding of our radiologists was compromised due to the delayed inclusion of the controls caused by obstacles related to the COVID-19 pandemic. The assessing radiologists were aware of the recruitment issues, which could have introduced bias to the radiological assessment. Second, the small sample size increases the risk of sampling bias and limits generalizability, although patient demographics and symptoms align well with previous studies of MN [40,41]. Third, the absence of a reference standard limits the possibility of validating our results and testing the diagnostic accuracy. Since our purpose was to assess the incidence of IMB and MN in a population with intermetatarsal pain, and not among surgical candidates, this could not be obtained due to ethical reasons. Additionally, inter-reader reliability was not assessed but should be considered in future studies, especially given the complexity of differentiation documented in our study. Lastly, the time interval between scans was extended in some cases, primarily among the controls and one patient. Regarding the controls, we find it unlikely that they affected the results, as they remained pain-free. In the case of one of our patients, the interval was 3 months due to circumstances unrelated to the patient. Given the mean pain duration of 28 months, persistent pain in the patient, and the fact that no interventions were initiated during the intermediate period, we consider it unlikely that this delay would have affected the imaging findings. The remaining patients had both scans performed within 0–34 days. Based on these considerations and our small sample size, these cases are retained in our study population.

## 5. Conclusions

Based on our MRI results, we conclude that IMB is a frequent cause of intermetatarsal pain and should be included in the differential diagnoses when evaluating patients with web space pain. US is valuable in detecting pathological changes in the intermetatarsal space, but differentiation between MN and IMB requires caution and an understanding of both conditions. The normal intermetatarsal bursa is visible as a non-expansive hypoechoic mass, which emphasizes the need for clinical correlation and thorough evaluation. Increased diagnostic attention to IMB may improve the management and treatment of patients with intermetatarsal pain.

## Figures and Tables

**Figure 1 diagnostics-15-01339-f001:**
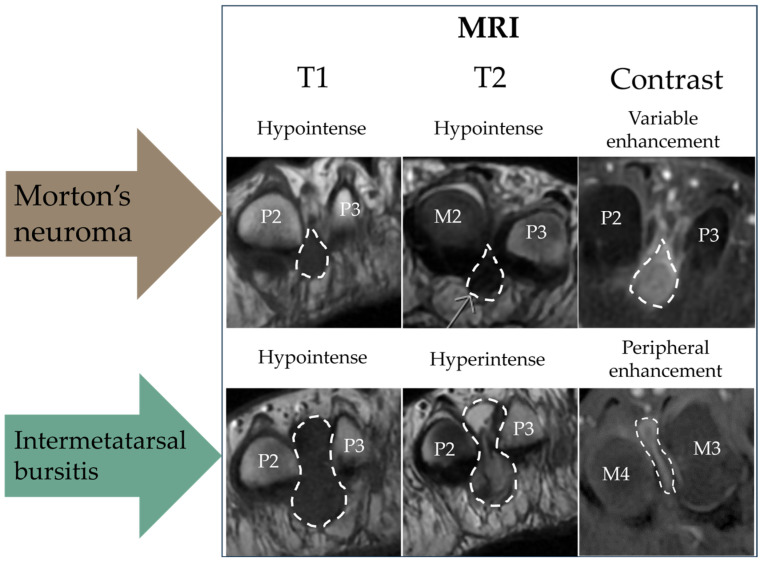
Schematic diagram of the magnetic resonance imaging (MRI) method to differentiate between Morton’s neuroma (MN) and intermetatarsal bursitis (IMB). All scans are in the transverse plane. P = phalanx; M = metatarsal. Please note that the images were selected to illustrate typical features of each sequence and originate from different patients.

**Figure 2 diagnostics-15-01339-f002:**
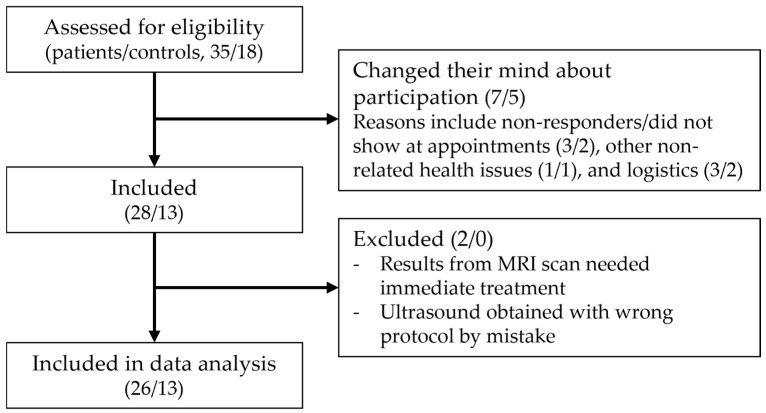
Flowchart of the inclusion process (patients/controls).

**Figure 3 diagnostics-15-01339-f003:**
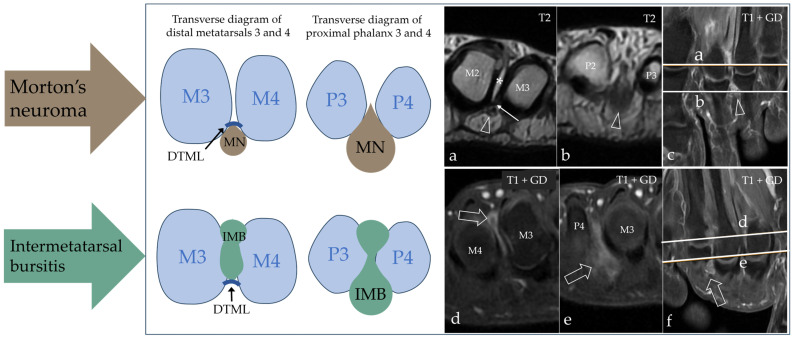
Schematic diagram of the anatomic landmarks used in the differentiation between MN and IMB on MRI. Both conditions have a proximal extent into the space between the metatarsal heads; here, they are separated by the deep transverse metatarsal ligament (DTML) in the dorso-plantar direction. The space is narrow and limits the information from radiological evaluation as demonstrated in (**a**) (where the open arrowhead points at MN, * marks fluid in the intermetatarsal bursa, and the thin arrow points at the DTML) and (**d**) (where the open arrow points at IMB); note the plantar location of MN and the dorsal location of IMB. More distally at the level of the metatarsophalangeal joints and between the proximal phalanges, we obtain the best view of the entities ((**b**), where the open arrowhead points at MN, and (**e**), where the open arrow points at IMB). Without the DTML to separate them, they have overlapping anatomic locations. The references for the proximal–distal location of the axial scans are shown on (**c**) for MN and (**f**) for IMB. M = metatarsus; P = phalanx.

**Figure 4 diagnostics-15-01339-f004:**
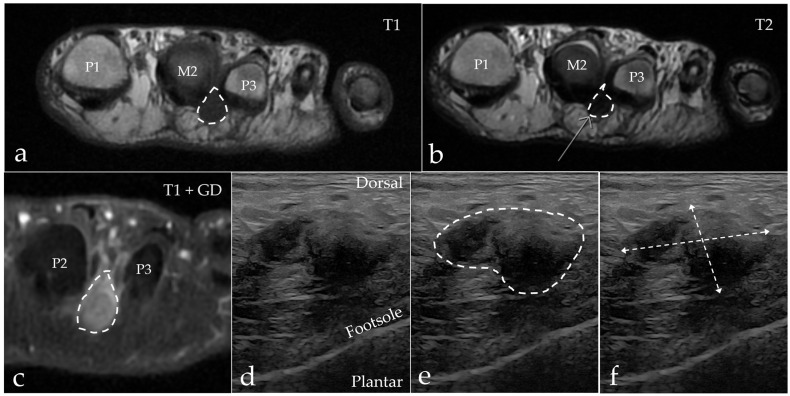
A 63-year-old woman with MN (marked with a dotted line) in the second web space on the left foot. (**a**) Axial MRI T1-weighted FSE image showing a hypointense process extending plantar. (**b**) Axial MRI T2-weighted FSE image showing the process with a hypointense appearance. (**c**) Axial MRI T1-weighted FSE Flex contrast-enhanced image showing the contrast enhancement of the process. (**d**,**e**) Sagittal ultrasound of the second web space in the same patient. Note the expansive appearance of the hypoechoic mass as marked with the dashed line. (**f**) The process measures 11.5 mm × 19.0 mm. M = metatarsus, P = phalanx.

**Figure 5 diagnostics-15-01339-f005:**
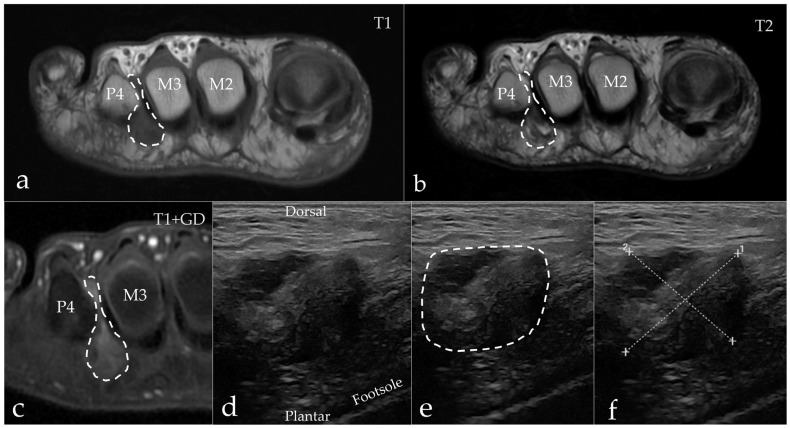
A 57-year-old man diagnosed with IMB (arrows) in the third web space on the right foot. (**a**) Axial MRI T1-weighted FSE image showing a hypointense process extending both dorsally and plantarly. (**b**) Axial MRI T2-weighted FSE image showing the process inhomogeneous, but predominantly hyperintense. (**c**) Axial MRI T1-weighted FSE Flex contrast-enhanced image showing the peripheral enhancement of the process. (**d**,**e**) Sagittal ultrasound of the third web space in the same patient. Note the expansive appearance of the hypoechoic mass as marked with the dashed line. (**f**) The process measures 18.7 mm × 17.1 mm. M = metatarsus, P = phalanx.

**Figure 6 diagnostics-15-01339-f006:**
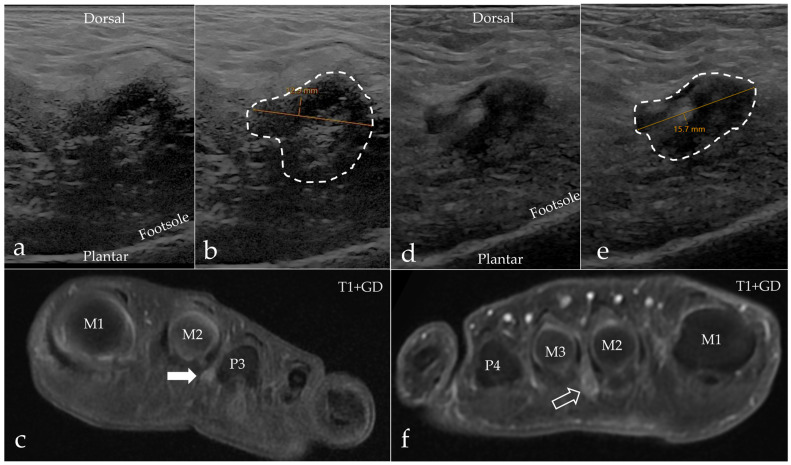
Examples of asymptomatic web space findings from the control group. (**a**–**c**) A 27-year-old female’s second web space, left foot, with a visible but nonpathological bursa on sagittal ultrasound measuring 12.9 mm (**a**,**b**) and axial MRI with reported IMB ((**c**), white arrow). (**d**–**f**) A 52-year-old female’s second web space, right foot, diagnosed with IMB on both sagittal ultrasound (**d**) and axial MRI ((**f**), open arrow). The hypoechoic mass measures 15.7 mm (**e**).

**Figure 7 diagnostics-15-01339-f007:**
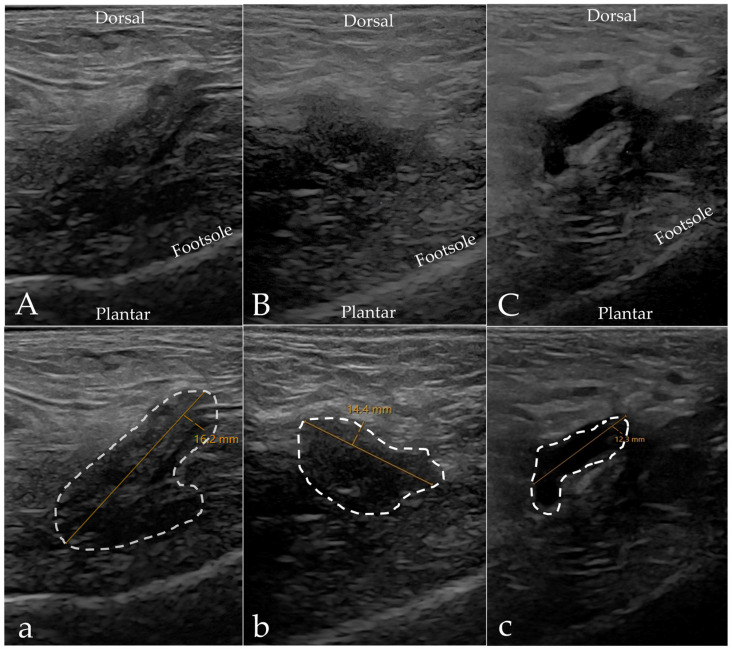
Examples of visible intermetatarsal bursas on ultrasound in our control group. (**A**,**a**) A 47-year-old woman’s third web space, left foot; ultrasound reported IMB measuring 16.2 mm. (**B**,**b**) A 65-year-old man’s second web space, right foot; ultrasound reported a visible but normal bursa measuring 14.4 mm. (**C**,**c**) A 53-year-old woman’s second web space, left foot; ultrasound reported a visible but normal bursa measuring 12.3 mm. All scans were obtained in the sagittal plane in a dorso-plantar direction. Left is proximal and right is distal. Note the presence of concave sides and a non-expansive appearance of the hypoechoic process, as marked with the dashed line. The MRI scans for all three patients were normal.

**Figure 8 diagnostics-15-01339-f008:**
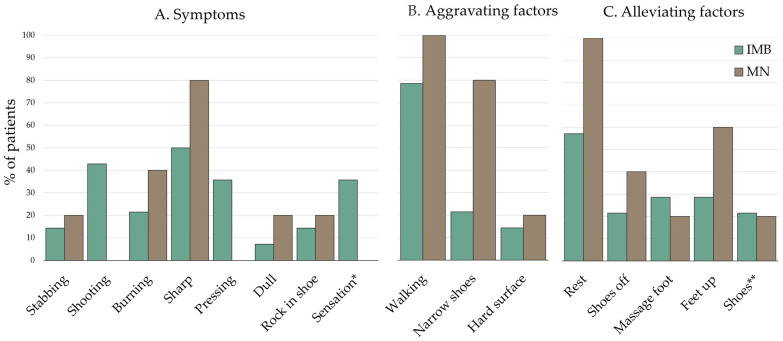
Frequency of (**A**) symptoms, (**B**) alleviating factors, and (**C**) aggravating factors, as reported by the patients. * “Sensation” refers to altered feeling in the foot, e.g., feels like walking on cottonwool, cold feeling, walking on gravel, etc. ** “Shoes” refers to the use of a specific shoe or insole. Subgrouping was based on MRI diagnosis.

**Figure 9 diagnostics-15-01339-f009:**
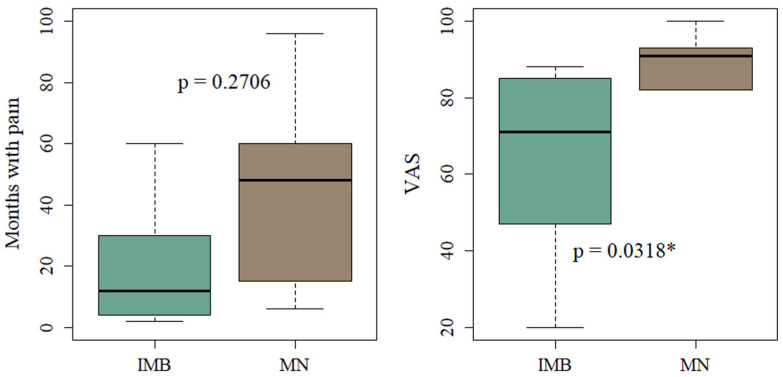
Boxplots of duration of pain and VAS score of patients with IMB and MN according to MRI diagnosis. * Indicates significant values (*p* < 0.05).

**Table 1 diagnostics-15-01339-t001:** Overview of patients’ clinical information obtained at the time of inclusion.

Mean Pain Duration	Affected Foot R/L	Affected Web Space (Subjective)	Affected Sensation in Adjacent Toes	Triggering Trauma	Mean VAS Score at Inclusion
29 months (range 2–96)	16:10	2nd: 7 (27%) 3rd: 6 (23%) 2nd + 3rd: 8 (31%) Non-specific: 5 (19%)	17 (65%)	5 (19%)	67 (range 20–100) *

* VAS score was missing for one patient.

**Table 2 diagnostics-15-01339-t002:** Results of radiological evaluation.

	Patients		Controls	
	**MRI**	**US**	**MRI**	**US**
Intermetatarsal bursitis	53.9% (14/26)	96.2% (25/26)	23.1% (3/13)	30.8% (4/13)
Morton’s neuroma	19.2% (5/26)	0% (0/26)	7.7% (1/13)	0% (0/13)
Normal scan	19.2% (5/26)	3.8% (1/26)	61.5% (8/13)	69.2% (9/13)
Other	7.7% (2/26)	0% (0/26)	7.7% (1/13)	0% (0/13)

## Data Availability

The raw data supporting the conclusions of this article will be made available by the authors upon request to SBL. The imaging material is not available due to privacy reasons.

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
