# Peer review of "Morton’s Neuroma or Intermetatarsal Bursitis—A Prospective Diagnostic Study of Intermetatarsal Pain"

_diagnostics, 2025, doi:10.3390/diagnostics15111339_

Round 1
Reviewer 1 Report (New Reviewer)
Comments and Suggestions for Authors
In the review of "Morton’s Neuroma or Intermetatarsal Bursitis - A Prospective Diagnos-tic Study of Intermetatarsal Pain,"
There are some major and minor issues that need to be addressed.
It is an interesting topic to help clinicians and imagers. I think if authors could clarify the statistically significant clinical findings and imaging findings in one table, that would be a great help for readers.
If, for ultrasound MRI findings, authors would have two blind readers for observation documentation, that would be great. ICC with CI is also helpful for readers.
Change the font of images to the font of text and also put the name of the image on the left lower corner.
It seems the author was not able to distinguish IMB from MN on imaging. Please provide a few characteristic imaging findings for each entity.
Table 1. Revise the table by rounding numbers without decimals. Transpose the table to show long titles on the column in a better way.
Figure 2. Define the X and Y axes here. Authors also can show their confidence intervals on the bar plots. Remove all patients, as this is not helpful in any way. Increase the size of the bars and also the X and Y axes with their labels to be the same as the text of the paper.
Figures 3 and 4. Change the colors of the arrows to white or red. Start the captions with the sex and age of the subjects. Put the labels on the left lower corner with the same font of text, which seems to be Times…
Figure 4. Remove the ultrasound measurement in F and annotate the image by PowerPoint.
Figure 5. Explain each image separately with a standard of age and sex introduction. Change the Image annotation to the left lower corner to Times font
Figure 6. The caption is not suitable. Define it based on the same standard as above suggestions
Figure 7. Authors need to show p-values over the boxes. Please use different colors for MN and IMB. Remove the outliers from the images.
Authors need to provide a table of MRI features of both IMB and MN to illustrate the characteristic findings. In addition, please compare the anatomical landmarks for each of these entities. I could not find any imaging marker to discriminate between these two pathologies.
The first paragraph of the discussion needs to be a summary of the results without numbers.
Each paragraph of discussion should discuss the result.
It is necessary for discussion to have a structure. First sentence of each paragraph would be a one sentence summary of one finding of study, then discuss and explain that finding in detail, and next add a literature review.
Are these two pathologies look exactly the same on US and MRI? Make it clear.
Comments on the Quality of English Language
It would be better to revise the captions and also statistical images to the standard of research papers.
Please describe imaging findings with an attention to the structural reporting in the radiology.
Author Response
Response to Reviewer 1
Thank you very much for conducting a thorough review and providing valuable input on our manuscript. Your feedback is greatly appreciated. We have addressed your concerns and suggestions under each comment and incorporated them into an improved version of the manuscript.
Open Review
( ) I would not like to sign my review report
(x) I would like to sign my review report
Quality of English Language
(x) The English could be improved to more clearly express the research.
( ) The English is fine and does not require any improvement.
Yes |
Can be improved |
Must be improved |
Not applicable |
|
Does the introduction provide sufficient background and include all relevant references? |
( ) |
(x) |
( ) |
( ) |
Is the research design appropriate? |
( ) |
(x) |
( ) |
( ) |
Are the methods adequately described? |
( ) |
(x) |
( ) |
( ) |
Are the results clearly presented? |
( ) |
(x) |
( ) |
( ) |
Are the conclusions supported by the results? |
( ) |
(x) |
( ) |
( ) |
Comments and Suggestions for Authors
In the review of “Morton’s Neuroma or Intermetatarsal Bursitis – A prospective Diagnostic Study of Intermetatarsal Pain,”
There are some major and minor issues that need to be addressed.
Comment 1:
It is an interesting topic to help clinicians and imagers. I think if authors could clarify the statistically significant clinical findings and imaging findings in one table, that would be a great help for readers.
Response 1:
We have made a table of the imaging findings to provide a quick overview of our results (Table 2, page 6). In addition, we have moved the analysis of the symptoms down to section 3.3, Differences between MN and IMB, pages 9–10, so all analyses comparing MN and IMB are provided in the same section to improve readability.
Comment 2:
If, for ultrasound MRI findings, authors would have two blind readers for observation documentation, that would be great. ICC with CI is also helpful for readers.
Response 2:
Unfortunately, this was not the setup used in this study. We used consensus to resolve cases of doubt. However, given the complexity encountered, it would indeed have been a desirable approach. We have discussed this in the limitations section of the Discussion and intend to consider it for future studies:
“Additionally, inter-reader reliability was not assessed but should be considered in future studies, especially given the complexity of differentiation documented in our study”, page 12, lines 386–388.
Comment 3:
Change the font of images to the font of text and also put the name of the image on the left lower corner.
Response 3:
We have made these adjustments for all figures.
Comment 4
It seems the author was not able to distinguish IMB from MN on imaging. Please provide a few characteristic imaging findings for each entity.
Response 4:
To address this concern, we have added a figure to the Methods section to summarize the method used to differentiate MN and IMB—please see Figure 2, page 4.
Comment 5:
Table 1. Revise the table by rounding numbers without decimals. Transpose the table to show long titles on the column in a better way.
Response 5:
We have rounded the numbers to whole numbers. We have attempted to transpose the table, but Microsoft Word has caused some difficulties. Please see the pdf version if it does not appear fitted.
Comment 6:
Figure 2. Define the X and Y axes here. Authors also can show their confidence intervals on the bar plots. Remove all patients, as this is not helpful in any way. Increase the size of the bars and also the X and Y axes with their labels to be the same as the text of the paper.
Response 6:
We have moved the figure to Section 3.3, page 10, as mentioned in Response 1. We have updated the plot by removing the “all patients” category, changed it to represent the % of patients in each group for easier comparison, added a label to the y-axis, and updated the font and increased the size of the bars.
Additionally, we have analyzed the distribution of symptoms between IMB and MN patients, which showed no significant difference. We have not shown the confidence intervals as they are very wide due to our small sample size, an issue we expanded upon in the Discussion:
“the small sample size increases the risk of sampling bias and limits generalizability”, page 12, lines 380–382.
The analysis is provided in the Supplementary Material, Table S1, and the results are referred to in the text in Section 3.3:
“The dominant symptoms of the included patients were a sharp or shooting pain that was aggravated when walking and alleviated by rest. There was no statistical difference between the symptoms for patients with IMB compared to patients with MN (see analysis in supplementary material, Table S1)”, page 9, lines 264–268.
Comment 7:
Figures 3 and 4. Change the colors of the arrows to white or red. Start the captions with the sex and age of the subjects. Put the labels on the left lower corner with the same font of text, which seems to be Times…
Response 7:
Figures 3 and 4 (now 4 and 5, page 7) have been updated accordingly.
Comment 8:
Figure 4. Remove the ultrasound measurement in F and annotate the image by PowerPoint.
Response 8:
The ultrasound measurement was removed and annotated accordingly; see Figure 5, page 7.
Comment 9:
Figure 5. Explain each image separately with a standard of age and sex introduction. Change the Image annotation to the left lower corner to Times font
Response 9:
Figures 5 (now 7, page 9) has been updated accordingly.
Comment 10:
Figure 6. The caption is not suitable. Define it based on the same standard as above suggestions
Response 10:
Figures 6, page 8, has been updated accordingly.
Comment 11:
Figure 7. Authors need to show p-values over the boxes. Please use different colors for MN and IMB. Remove the outliers from the images.
Response 11:
Figure 7 (now 9, page 10) now shows p-values and has different colors. The outliers have been removed.
Comment 12:
Authors need to provide a table of MRI features of both IMB and MN to illustrate the characteristic findings. In addition, please compare the anatomical landmarks for each of these entities. I could not find any imaging marker to discriminate between these two pathologies.
Response 12:
We have added Figure 3 on page 6, which details the anatomical landmarks combined with the MRI features that differ IMB from MN.
Comment 13:
The first paragraph of the discussion needs to be a summary of the results without numbers.
Response 13:
We have revised the first paragraph of the discussion according to your comment:
“In this prospective study of patients with intermetatarsal pain, compared with a control group, radiological evaluation identified IMB or MN in most patients. MRI identified IMB in 54% of patients, making it the most frequent finding. MRI more often failed to identify a radiological explanation for the reported pain, but US was unable to differentiate MN from IMB. There were no differences in the symptoms between IMB and MN; however, patients with MN had a longer pain history on average and significantly higher VAS scores than those with IMB. In the control group, asymptomatic web space pathology was common on imaging and present in nearly one-third of the individuals. Additionally, we found that the normal intermetatarsal bursa is visible on US.” Pages 10-11, lines 283–291.
Comment 14 and 15:
Each paragraph of discussion should discuss the result.
It is necessary for discussion to have a structure. First sentence of each paragraph would be a one sentence summary of one finding of study, then discuss and explain that finding in detail, and next add a literature review.
Response 14 and 15:
We have made major revisions to the Discussion to fit the described structure.
Comment 16:
Are these two pathologies look exactly the same on US and MRI? Make it clear.
Response 16:
We have made it more clear how we distinguished between the two conditions with the addition of Figure 2, page 4, and Figure 3, page 6, and the issues with the ultrasound in Figure S1, Supplementary Materials.
Comments on the Quality of English Language
It would be better to revise the captions and also statistical images to the standard of research papers.
Please describe imaging findings with an attention to the structural reporting in the radiology.
Response:
With the help of your detailed input for improvement of figures and tables, the captions are now uniform, present structural base fitting for imaging, and contain information required in an academic manuscript.

Reviewer 2 Report (New Reviewer)
Comments and Suggestions for Authors
Well written paper and study
however cohort is too small
diagnostic criteria of IMB and Mn on mri I am afraid is incorrect . For example figure 4 lesion extends plantar to deep intermetatrsal ligament hence is Morton’s neuron with IMB and not IMB alone
There is significant delay between mri and ultrasound - it’s well known that IMB and mm can change and hence this is an important favtor
ultrasound images should ideally have been reviewed by two radiologists and not one alone
Author Response
Response to Reviewer 2
Thank you very much for your feedback on our manuscript. We have responded your concerns under each comment and explored the methodological issues you point out more deeply in the Discussion.
Open Review
(x) I would not like to sign my review report
( ) I would like to sign my review report
Quality of English Language
( ) The English could be improved to more clearly express the research.
(x) The English is fine and does not require any improvement.
Yes |
Can be improved |
Must be improved |
Not applicable |
|
Does the introduction provide sufficient background and include all relevant references? |
(x) |
( ) |
( ) |
( ) |
Is the research design appropriate? |
( ) |
(x) |
( ) |
( ) |
Are the methods adequately described? |
( ) |
( ) |
(x) |
( ) |
Are the results clearly presented? |
( ) |
( ) |
(x) |
( ) |
Are the conclusions supported by the results? |
( ) |
(x) |
( ) |
( ) |
Comments and Suggestions for Authors
Well written paper and study /Thank you.
Comment 1:
however cohort is too small
Response 1:
We have addressed this concern in the Methods section and discussed the possible implications for our results:
“Second, the small sample size increases the risk of sampling bias and limits generalizability, although patient demographics and symptoms align well with previous studies of MN”, page 12, lines 380–382.
Comment 2:
diagnostic criteria of IMB and Mn on mri I am afraid is incorrect . For example figure 4 lesion extends plantar to deep intermetatrsal ligament hence is Morton’s neuron with IMB and not IMB alone
Response 2:
We realize we have been too vague in our description of our differentiation between MN and IMB. Therefore, we have added Figure 2, page 4, and Figure 3, page 6, to better visualize the applied methods and the anatomic considerations that aid in differentiation.
Comment 3:
There is significant delay between mri and ultrasound - it’s well known that IMB and mm can change and hence this is an important favtor
Response 3:
We have addressed this concern more thoroughly in the section of the Discussion covering limitations:
“the time interval between scans was extended in some cases, primarily among the controls and one patient. Regarding the controls, we find it unlikely to have affected the results, as they remained pain-free. In one case of our patients, the interval was three months due to circumstances unrelated to the patient. Given the mean pain duration of 28 months, persistent pain in the patient, and that no interventions were initiated during the intermediate period, we consider it unlikely that this delay would have affected the imaging findings. The remaining patients had both scans performed within 0-34 days. Based on these considerations and our small sample size, these cases are retained in our study population.” Pages 12–13, lines 388–396.
Additionally, we have added a graph to the Supplement Materials, Figure S2 (b), visualizing the difference between patients with IMB and MN. There is no significant difference between the patient subgroups.
Comment 4:
ultrasound images should ideally have been reviewed by two radiologists and not one alone
Response 4:
Yes, we agree that the study could have benefited substantially from this. Unfortunately, this was not the setup we used, but given the complexity we experienced, it would have been desirable. We have discussed this in the limitations section of the Discussion and its consideration for future studies:
“Additionally, inter-reader reliability was not assessed but should be considered in future studies, especially given the complexity of differentiation documented in our study” page 12, lines 386–388.

Round 2
Reviewer 2 Report (New Reviewer)
Comments and Suggestions for Authors
Thanks for revising paper
looks good
This manuscript is a resubmission of an earlier submission. The following is a list of the peer review reports and author responses from that submission.
Round 1
Reviewer 1 Report
Comments and Suggestions for Authors
Dear Authors,
It is an interesting topic and the article is well described.
However, the introduction can be improved, the clinical tests must be include in the introduction and also the symptoms must be detailed.
Methods: the clinical test for neuroma was done? I think it could be important to explain it.
thank you
also the type of footwear and the profession and sports, could be interesting to compare.
The discussion must be more detailed, with more recent articles to compare wiht the results.
Reviewer 2 Report
Comments and Suggestions for Authors
The Authors aimed to investigate the incidence of Morton's Neuroma and Intermetatarsal bursytis in a prospective cohort of patients with pain in a foot web space compared.
Although the topic is interesting, the study is poorly designed.
The title is misleading.
Abstract is confusing.
Introduction: would further define MN and IMB.
The series is small. Also, the control cohort should be comparable. Also, intermetatrsal pain must be defined. How were controls included?
Interobserver reliability should be checked for US and MRI. Please define MN and IMB on both US and MRI. Which are characteristic criteria?
Were these findings observed in any control?
Fig 2. Cases with an interval between US and MRI >30 days must be excluded.
Discussion not relied on results
Comments on the Quality of English LanguageMany grammar and syntax errors.
